# Unraveling bilayer interfacial features and their effects in polar polymer nanocomposites

Xinhui Li[1], Shan He[2], Yanda Jiang[1], Jian Wang[1], Yi Yu[3], Xiaofei Liu[1], Feng Zhu[4], Yimei Xie[1], Youyong Li [3], Cheng Ma [4], Zhonghui Shen [1], Baowen Li[1], Yang Shen [2], Xin Zhang [1,5] ✉, Shujun Zhang [6] ✉ & Ce-Wen Nan [2] ✉

Polymer nanocomposites with nanoparticles dispersed in polymer matrices have attracted extensive attention due to their significantly improved overall performance, in which the nanoparticle-polymer interface plays a key role. Understanding the structures and properties of the interfacial region, however, remains a major challenge for polymer nanocomposites. Here, we directly observe the presence of two interfacial polymer layers around a nanoparticle in polar polymers, i.e., an inner bound polar layer (~10 nm thick) with aligned dipoles and an outer polar layer (over 100 nm thick) with randomly oriented dipoles. Our results reveal that the impacts of the local nanoparticle surface potential and interparticle distance on molecular dipoles induce interfacial polymer layers with different polar molecular conformations from the bulk polymer. The bilayer interfacial features lead to an exceptional enhancement in polarity-related properties of polymer nanocomposites at ultralow nanoparticle loadings. By maximizing the contribution of inner bound polar layer via a nanolamination design, we achieve an ultrahigh dielectric energy storage density of 86 J/cm³, far superior to state-of-the-art polymers and nanocomposites.

Polymer nanocomposites dispersed with nanoparticles exhibit significantly improved mechanical, electrical and thermal properties compared to pure polymers[1–6]. It is generally believed that the emerging nanoparticle-polymer interface is essential for achieving their property enhancements[7–10]. The conformation of the interfacial polymers (e.g., segregation of chain ends, the density of polymer segments and orientation of the side groups) often changes near the surface of solid nanoparticles[11,12], inducing a deviation in properties compared to the bulk polymer in terms of glass transition temperature $T_g$, mobility and viscosity due to the existence of nanoconfinement and surface effects[13–16]. Although the presence of interfacial polymers provides an opportunity to tailor the properties of nanocomposites, the interfacial effects in polymer nanocomposites remain unclear, as direct characterization of the nanoscale interfacial regions buried around tough nanoparticles in soft polymer matrix is a great challenge.

[1]State Key Laboratory of Advanced Technology for Materials Synthesis and Processing, Center of Smart Materials and Devices, Wuhan University of Technology, Wuhan, Hubei, China. [2]School of Materials Science and Engineering, State Key Lab of New Ceramics and Fine Processing, Tsinghua University, Beijing, China. [3]Institute of Functional Nano and Soft Materials, Jiangsu Key Laboratory for Carbon-Based Functional Materials & Devices, Soochow University, Suzhou, China. [4]Division of Nanomaterials & Chemistry, Hefei National Research Center for Physical Sciences at the Microscale, CAS Key Laboratory of Materials for Energy Conversion, Department of Materials Science and Engineering, University of Science and Technology of China, Hefei, Anhui, China. [5]International School of Materials Science and Engineering, Wuhan University of Technology, Wuhan, Hubei, China. [6]Institute for Superconducting and Electronic Materials, Australian Institute of Innovative Materials, University of Wollongong, Wollongong, NSW, Australia. ✉e-mail: zhang-xin@whut.edu.cn; shujun@uow.edu.au; cwnan@tsinghua.edu.cn

To understand the interfacial effects on nanocomposite properties, a few theoretical models have been proposed, in which the interfacial polymer was considered to consist of different layers including the tightly bound layer and the loosely bonded layer[17–22]. These models describing interfacial polymers with assumed nanoscale interfacial layers can explain some phenomena in nanocomposites, but the current interface models are abstract assumptions where the existence of these interfacial layers has not been directly demonstrated. Experimentally, a model nanocomposite was designed, in which the polymer thin film was confined between two silica slides, to understand the effect of interfacial polymer layers[23]. The quantitative correlation between thin-film thickness and interparticle distance in the nanocomposite provides a better understanding of the effect of interfacial polymer layers on $T_g$ in nanocomposites filled with silica nanoparticles[23,24]. More recently, direct investigation of the polymer-nanoparticle interfaces has been enabled by scanning probe microscopy techniques[25–31]. The surface potential mapping across the nanoparticle-polymer interfaces allows detection of the interfacial layer with higher electrical polarization compared to the bulk polymer[25], while the interface chemical mapping in ferroelectric polymer nanocomposites reveals that the interfacial regions around the nanoparticles are highly inhomogeneous with conformational disorders[26]. These recent attempts have provided direct experimental evidence that there exists an interfacial region with distinct local polarization and conformation from the bulk polymer, but it remains puzzling whether the hypothesized different interfacial polymer layers in theoretical models actually exist and how these layers are spatially distributed in terms of size, structure and properties. Of particular importance is the fact that the interfacial region around one nanoparticle in a polymer nanocomposite will be severely affected by its neighboring particles, since the interparticle distance decreases with increasing nanoparticle loading. Uncovering the possible influence of adjacent particles on interfacial polymers is another challenge for a comprehensive understanding of the interfacial effect in polymer nanocomposites.

Here, by protruding half of the nanoparticles out of the polar polymer matrix to facilitate the localization and detection of the buried interface region, we directly observe the interfacial polymer region surrounding the nanoparticle, which consists of two interfacial polymer layers (i.e., inner bound polar layer of about 10 nm thick and outer polar layer of over 100 nm thick) with different polar molecular conformation from the bulk polymer. By combining simulations and experiments, we illustrate that both the nanoparticle-polymer interaction and the interparticle distance affect the formation of the interfacial polymer layers, which leads to an exceptional enhancement of polarity-related properties in dilute polymer nanocomposites. To maximize the effect of the inner bound polar layer, we design a nanolaminated nanocomposite with the polymer thin film confined between two oxide nanolayers, and achieve an ultrahigh dielectric energy storage density of the nanocomposite. Our observations provide an intuitive understanding of the interfacial effect in polymer nanocomposites and an effective strategy to tailor the polarity-related properties of nanocomposites via interface engineering.

## Results and Discussion
### Direct observation of interfacial features around a nanoparticle
For demonstration, we choose poly(vinylidene fluoride) (PVDF, the most common ferroelectric polymer) as a matrix and TiO$_2$ nanoparticles as the nanofillers. PVDF and its copolymers are typical polar polymers with excellent dielectric and piezoelectric properties[32,33]. To facilitate the localization and detection of the buried interface area from the nanocomposite surface, the PVDF-TiO$_2$ nanocomposites were prepared via a carefully treated spin-coating process (Methods and Fig. 1a) to make the nanoparticles protrude out of the polymer matrix. As shown in Fig. 1a, the thickness ($t$) of the PVDF polymer film is

deliberately reduced to a value smaller than the diameter of nanoparticle ($D$), enabling the nanoparticles to protrude from the polymer matrix. In the experimental process, a pure PVDF polymer layer was first spin-coated onto silicon wafer substrates to eliminate any potential surface effect originating from the silicon substrate. As shown in the scanning electron microscopy (SEM) image (Fig. 1b, c), a visible layer with a thickness of about 10 nm is retained on the nanoparticle surface protruding out of the polymer matrix, and this is an inner layer consisting of strongly adsorbed polymer chains, also known as the bound layer[34]. The high-angle annular dark-field (HADDF) image and the carbon mapping results (Fig. 1d) further illustrate that this inner bound interfacial layer possesses higher carbon content, suggesting a higher density of polymer segments in the bound layer than the bulk polymer. Such a similar high-density bound layer is also observed in PVDF nanocomposite with BaTiO$_3$ nanoparticles (Supplementary Fig. 1). The formation of the high-density layer is ascribed to the first arriving polymer molecules which are adsorbed onto the nanoparticle surface with a flat conformation[11].

We further investigated the chemical structure of the interfacial polymer by mapping the interfacial region with atomic force microscope infrared spectroscopy (AFM-IR), where an AFM tip was employed to locally detect the thermal expansion of the sample induced by infrared radiation (Fig. 1e). In PVDF matrix, polymer conformations with different polarities coexist, among which, the all-*trans* (TTTT, related to the polar $\beta$ phase), and three *trans* linked to a *gauche* (TTTG, related to the polar $\gamma$ phase) conformations have stronger polarity than the *trans-gauche* (TGTG, related to the non-polar $\alpha$ phase)[35]. The chemical mapping was conducted with an infrared laser at 840 cm$^{-1}$ and 766 cm$^{-1}$, which signify characteristic IR bands of the polar (TTTT/TTTG) and their counterpart non-polar (TGTG) conformation of PVDF chains, respectively[36]. The IR response of the interface region in the chemical maps is shown in Fig. 1f, meanwhile, the corresponding height image detected by the AFM tips is also given to show the morphology of the protruded nanoparticle. Interestingly, a larger conformation signal of 840 cm$^{-1}$ and a weaker conformation signal of 766 cm$^{-1}$ at the interface region are observed (Fig. 1f), which suggests the interfacial polymer predominantly exhibits a strong polar conformation. The thickness of the polar interfacial polymer layer is identified to be over 100 nm. The all-*trans* TTTT polar conformation of PVDF causes the dipoles to point from F to H atom due to the electronegativity difference. To investigate the dipole orientations, lateral and vertical piezoresponse force microscopy (PFM, Supplementary Figs. 2, 3) was employed[37,38]. As shown in Fig. 1g, the lateral PFM (L-PFM) characterizations were conducted along the in-plane x-direction and also the y-direction via 90° rotation of the sample, in both cases, the nanoparticle has a half-dark and half-bright contrast. In vertical PFM (V-PFM), however, the nanoparticle shows no contrast. By combining L-PFM and V-PFM phase images, the interface dipoles are determined to be aligned perpendicular to the nanoparticle surface as illustrated in Fig. 1e. In contrast, no phase signals can be detected from the pristine TiO$_2$ nanoparticle surface without the PVDF bound layer (Supplementary Fig. 4), which confirms that the orientation signal is originated from the bound polymer layer. The characterization of the presence of dipoles in the bound polymer layer also confirms that the inner 10 nm thick binding PVDF layer with high density also exhibits a TTTT polar conformation. According to these observations, we can reveal the interfacial polymers around the nanoparticle as shown in Fig. 1e., i.e., there are two interfacial polymer layers exhibiting polar conformations. The inner bound polar layer (about 10 nm thick) strongly interacts with the nanoparticle surface and exhibits aligned dipoles, while the outer polar layer weakly interacts with the nanoparticle surface and shows randomly orientated dipoles (over 100 nm thick). The observed bilayer interfacial features around a nanoparticle provide direct evidence supporting the assumptions of interfacial polymer layers in previous interface models[18–20,22].

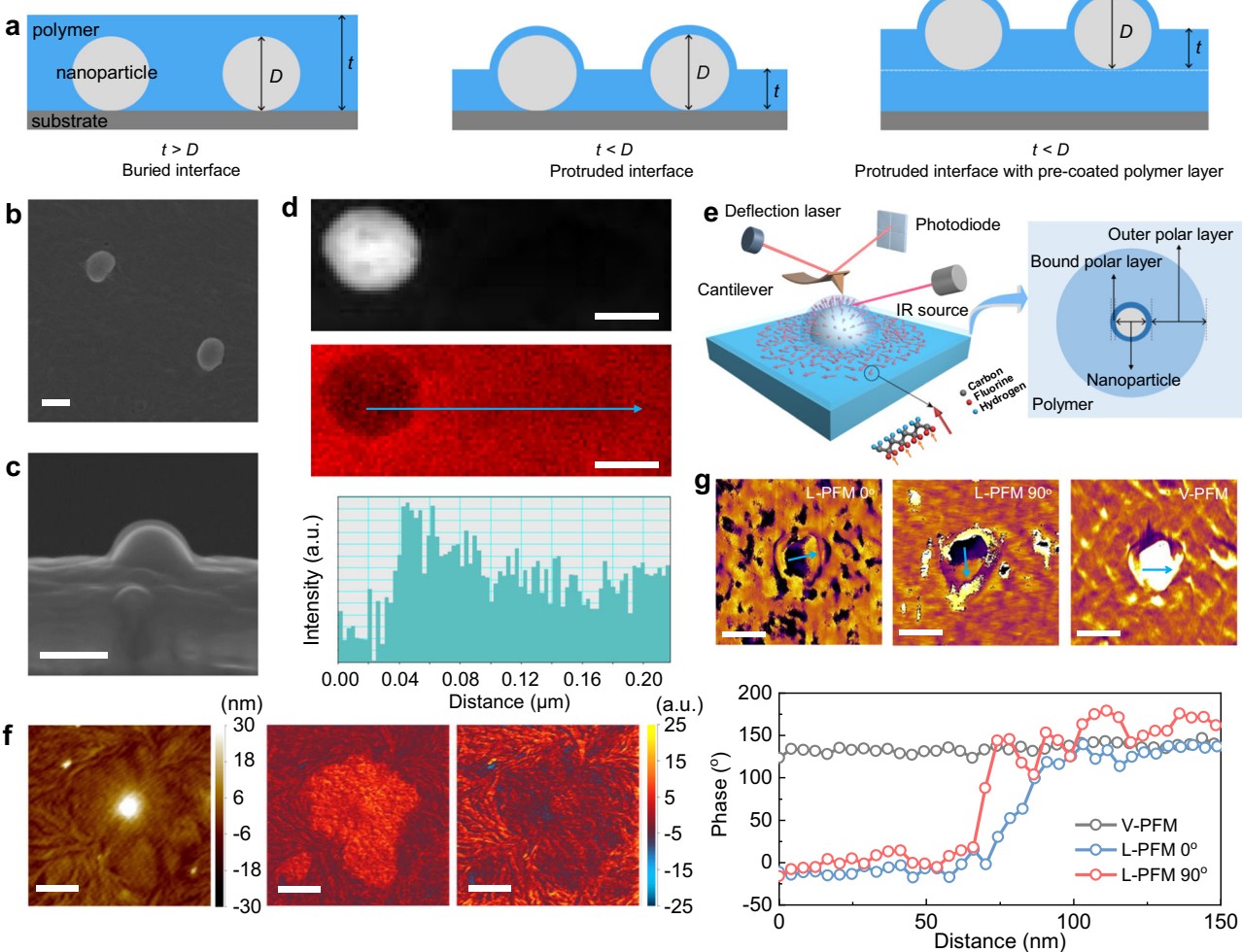

**Fig. 1 | Direct observation of the interfacial polymer around a nanoparticle in nanocomposites. a** Schematic illustration of the preparation of nanocomposite with protruded nanoparticles. *t* represents the thickness of the polymer film and *D* refers to the diameter of nanoparticle. **b**, **c** The surface (**b**) and cross-section (**c**) SEM images of the PVDF-TiO₂ nanocomposite with TiO₂ nanoparticle protruded out of the polymer matrix. The scale bar is 100 nm. **d** The HADDF image (up) of a TiO₂ nanoparticle embedded in the PVDF matrix, and the corresponding element mapping (middle) and line profile (blue) of carbon atom signal (bottom) near the interfacial region. The scale bar is 50 nm. **e** Schematic illustration of the AFM-IR measurement and the reconstructed dipole orientations in the interfacial region (left), and the interface model consisting of two interfacial polymer layers (right). **f** Simultaneously measured topography (left) and AFM-IR chemical maps with irradiation by a laser at 840 cm⁻¹ (middle) and 766 cm⁻¹ (right), respectively, of the interfacial region around the TiO₂ nanoparticle. The scale bar is 500 nm. **g** L-PFM, V-PFM phase images (up) and corresponding line profiles (bottom) showing the polarization components of the dipoles in the interface region. The scale bar is 200 nm.

## Interfacial polar regions in polymer nanocomposites

The distribution of such interfacial region in Fig. 1 is also dependent on the interparticle distance in the nanocomposites. As shown in Fig. 2a, b and Supplementary Fig. 5, three different cases are observed depending on the interparticle distance. First, the complete polar interfacial regions are formed around the isolated nanoparticles far away from the neighbors, also as described in Fig. 1. Second, when the interparticle distance is small, the overlap of the interfacial regions weakens the formation of the polar conformation. Third, when two nanoparticles are close enough to be interconnected, the interfacial regions also tend to be incorporated. The Fourier transform infrared spectroscopy (FTIR) analysis (Fig. 2c, d) further illustrates the variation of interfacial regions with the interparticle distance or nanoparticle loading in nanocomposites. According to the FTIR results (Fig. 2c), the ratios of absorbance intensities at 840 cm⁻¹/766 cm⁻¹ and 1279 cm⁻¹/766 cm⁻¹ are calculated, which signify the concentration ratios between polar TTTT and non-polar TGTG conformations (Fig. 2d). The content of the polar conformation continuously increases with increasing the volume fraction of the nanoparticles up to 0.35 vol.%,

and then decreases due to the reduced interparticle distance (Supplementary Fig. 6). Such a similar interfacial effect is also observed in the polymethyl methacrylate (PMMA)-TiO₂ nanocomposites, where the content of the mutual cis conformation of ester group and methoxy group (i.e., chain conformation with higher polarity in PMMA polymer) reaches a peak value at ~0.35 vol.% of TiO₂ nanoparticles (Supplementary Figs. 7, 8). Based on the observations in PVDF and PMMA-based nanocomposites, it is evident that the interfacial polar polymers are present in both crystalline and amorphous polar polymer nanocomposites. For crystalline polymers, such as PVDF, the as-formed interfacial polar polymers may undergo crystallization, leading to the development of interfacial polar crystalline phases, especially in the case of the *β* phase of PVDF.

In solution-processed polymer nanocomposites, the nanoparticles immersed in the solutions develop surface charges, and establish an electric double-layer containing mobile charges, dipoles and electronic polarizability to neutralize the surface potential.[19] The distributions of charges and potentials near the particle surface can be described by Poisson and Boltzmann equations with the help of the

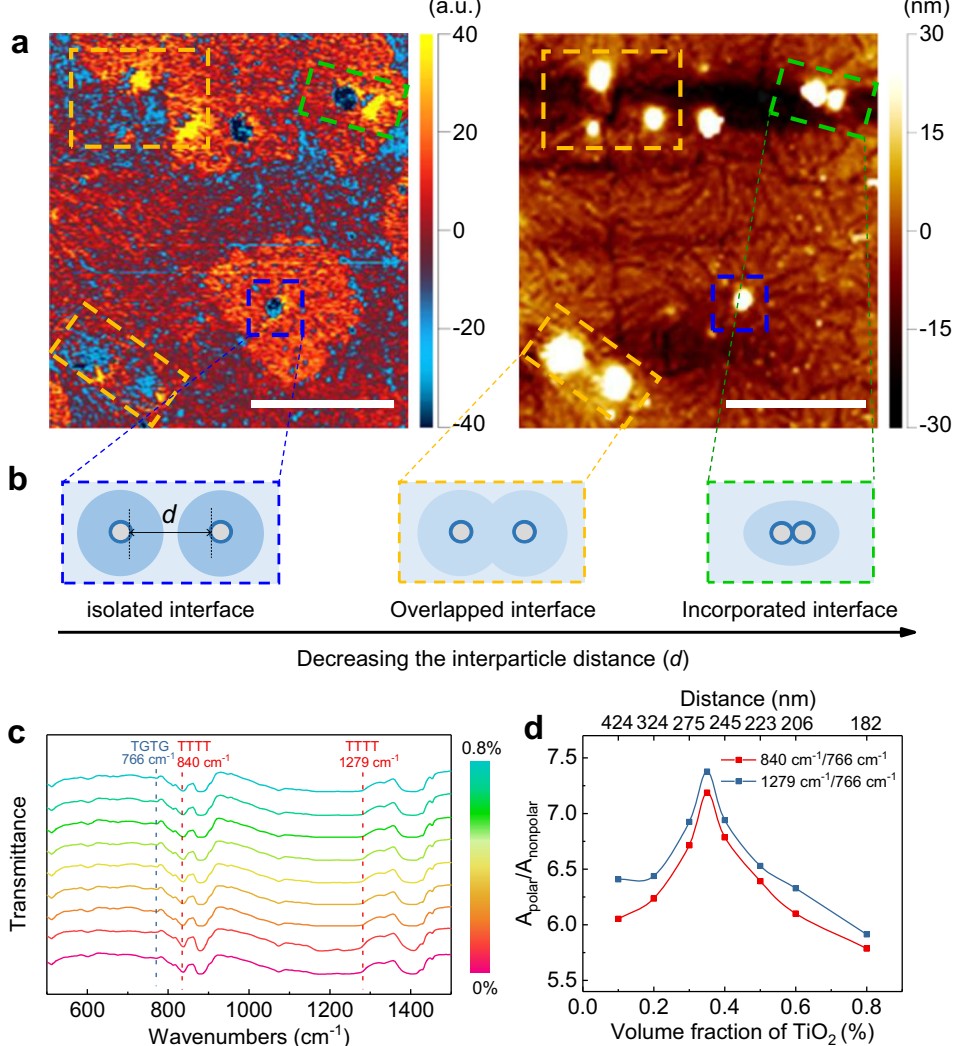

**Fig. 2 | Interfacial polar regions in nanocomposites. a** The AFM-IR chemical map with irradiation by a laser at 840 cm⁻¹ (left) and corresponding height image (right) of the PVDF-TiO₂ nanocomposite with different interparticle distances. The scale bar is 1.5 μm. **b** Schematic illustration of the interface model (Fig. 1e) considering the interparticle distance (*d*). The blue dashed rectangle shows the isolated interface, the orange dashed rectangle shows the overlapped interface, and the green dashed rectangle shows the incorporated interface. **c** FTIR spectra of the PVDF-TiO₂ nanocomposites with different volume fractions of nanoparticles. **d** The calculated ratio of absorbance intensities between polar ($A_{polar}$) and nonpolar ($A_{nonpolar}$) conformations at characteristic bands of 840 cm⁻¹, 1279 cm⁻¹ and 766 cm⁻¹ in the nanocomposites with different volume fractions of nanoparticles and different interparticle distances. The ratio of concentration (*n*) is in direct proportion to the ratio of absorbance intensity, i.e., $n_{840}/n_{766} = (a_{766}/a_{840})(A_{840}/A_{766})$, where *a* is the absorptivity.

classic Guoy-Chapman model.[39] The electric potential is believed to play a key role in driving the chain conformations of polar polymers via electrostatic interaction[40–42]. To further demonstrate the formation of polar polymer conformations at interfaces, molecular dynamics and phase-field simulations were performed. The molecular dynamic simulations show that the distribution of electric potential near the TiO₂ particle surface is the main enabler for the formation of polar conformations of PVDF. As shown in Fig. 3a, with increasing the electric potential of the particle surface, the content of TTTT polar conformation continuously increases while others decrease. Figure 3b shows the transition process of the PVDF polymer chain structure from the initial amorphous to the final stable polar conformation at different stages of electric field and time. The orientation of the generated dipoles tends to follow the direction of the surface potential, depending on the potential characteristics (negative or positive) of the nanoparticle surface in different matrix polymer solutions. It is noted that the experimentally observed thickness of the interface polar layers can be larger than the length of the Gouy-Chapman diffuse layer, because the as-formed polar conformations can continue to extend

the formation of the electric interfacial polar layers via long-range forces, such as Van der Waal's interaction and molecular dipolar interaction. Phase-field simulations, on the other hand, further demonstrate that an overlapped interface feature with a weakened interface polar conformation emerges, because the interface electric potential between the two nanoparticles decreases as they approach each other (Fig. 3c and Supplementary Fig. 9).

**Interfacial effect on the physical properties of nanocomposites**
The polar conformation of PVDF favours higher polarity-related dielectric and piezoelectric properties, and thus the two interfacial polar layers are expected to improve the properties of polymer nanocomposites with appropriate average interparticle distance (or nanoparticle volume fraction). As shown in Fig. 4a and Supplementary Fig. 10, the dielectric constant ($\varepsilon_r$) of the PVDF-TiO₂ nanocomposites as a function of the nanoparticle volume fraction in the PVDF-TiO₂ nanocomposites shows a similar trend to the polar polymer conformation content (Fig. 2d), where the $\varepsilon_r$ reaches the highest value of 9.5 at 0.35 vol.%, with a ~ 35% enhancement comparing to the pristine

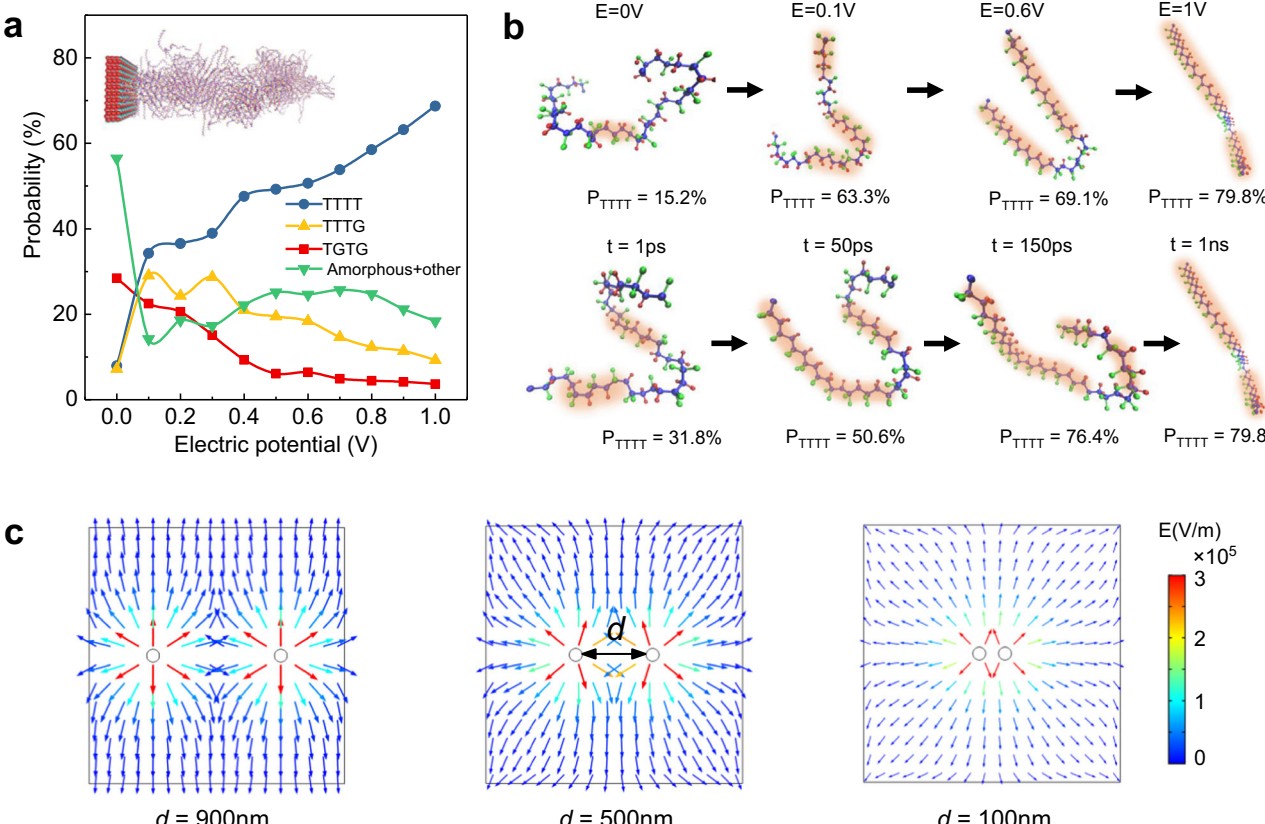

**Fig. 3 | Molecular dynamic and phase-field simulations of the interfacial polar polymers. a** The simulated content variations of different polymer conformations in PVDF under the driving electric potential, where the inset is the constructed interface model for molecular dynamic simulations containing a TiO$_2$ slab and interfacial PVDF polymers. **b** The views of the conformations of a single PVDF chain at different stages of time (t) and electric field (E) during the conformation transition process. The TTTT conformations are marked with orange shadow, P$_{TTTT}$ represents the probability of the TTTT conformation within the PVDF chain. **c** The phase-field simulated distributions of electric potential between two adjacent particles with different interparticle distances (d).

PVDF. This enhancement due to the bilayer interfacial effect goes far beyond the predictions by the existing rules of mixture (Supplementary Fig. 11). A similar dielectric enhancement is also observed in various nanocomposites with different polymer matrices such as polyetherimide (PEI) and PMMA (Fig. 4b, Supplementary Fig. 11), as well as in nanocomposites containing various nanoparticles including BaTiO$_3$. Despite BaTiO$_3$ exhibiting higher $\varepsilon_r$ than TiO$_2$, both PVDF-BaTiO$_3$ and PVDF-TiO$_2$ nanocomposites demonstrate comparable dielectric enhancement and polar conformations across the volume fraction range of 0.1–1 vol.%, indicating that the intrinsic $\varepsilon_r$ of the nanoparticles may have little impact on the formation of the polar interfacial layers (Supplementary Fig. 12). It is observed that the diameter of the nanoparticles has a close correlation with the occurrence of peak values of $\varepsilon_r$, where larger particle sizes necessitate a higher volume fraction of nanoparticles to achieve the peak value of the $\varepsilon_r$. For instance, the required volume fraction for achieving the highest $\varepsilon_r$ shifts from 0.2 vol.% to 0.7 vol.% as the diameter of the TiO$_2$ increases from 40 nm to 80 nm (Supplementary Fig. 13). These observations are consistent with previously reported polymer nanocomposites whose dielectric constants exhibit peak values over the dilute nanoparticle volume fraction of 0.2–0.9 vol%[43,44], indicating the universality of the interfacial polar layer effect in polymer nanocomposites. It should be noted that the peak enhancement of the $\varepsilon_r$ in PVDF-TiO$_2$ nanocomposites no longer exists as the temperature increases to 170 °C (Fig. 4a and Supplementary Fig. 14), which exceeds the melting temperature of PVDF (Supplementary Fig. 15). Above the melting temperature, the polymer chains become highly mobile which makes the formation of interfacial polar conformations difficult. In comparison, the dielectric enhancement in PEI-TiO$_2$ nanocomposites remains even up to 200 °C (Fig. 4b and Supplementary Fig. 14), since PEI possesses a higher melting temperature of above 250 °C.

The dielectric enhancement in polymer nanocomposites at the dilute concentration finds great potential for remarkably improved energy storage performance. Electrical energy storage is one of the most important applications of dielectric polymers[45], where both high electrical displacement (D, which is determined by $D = \varepsilon_r \varepsilon_0 E$, $\varepsilon_0$ is the vacuum dielectric permittivity and E is the applied electric field) and high breakdown strength ($E_b$, the critical value of applied electric field that a dielectric can withstand) are requisite to achieve a high energy density. The increase of $D(\varepsilon_r)$ in polymer nanocomposites typically leads to a reduction in $E_b$[46], while the rise in $\varepsilon_r$ resulting from polar polymer conformations maintains a high level of $E_b$ for the dilute nanocomposites. As illustrated in Supplementary Figs. 16, 17, incorporating 0.35 vol.% of TiO$_2$ nanoparticles produces over 40% enhancement in D, while simultaneously preserving a high $E_b$ of 630 kV/mm. The energy storage density and efficiency of the nanocomposites are calculated from the D-E loops (Supplementary Fig. 18), where dilute nanocomposite with 0.35 vol.% TiO$_2$ nanoparticles produces the highest energy density of 27 J/cm$^3$, which is 35% enhancement over pristine PVDF, while the energy efficiency remains on the order of 80% with the nanoparticle volume fraction ranging from 0 to 2 vol.% (Fig. 4c and Supplementary Fig. 19). The energy density and efficiency show no signs of degradation after 10$^6$ consecutive cycles, indicating excellent reliability and stability for nanocomposite with 0.35 vol.% nanoparticle addition (Supplementary Fig. 20).

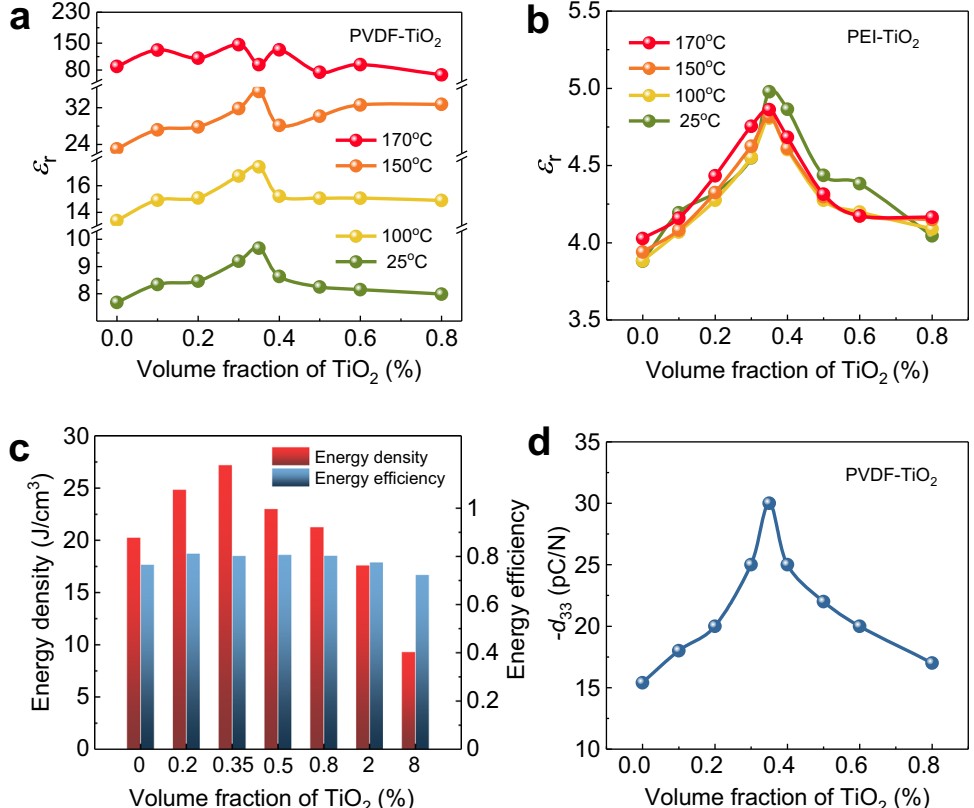

**Fig. 4 | Interfacial effect on the overall properties of the nanocomposites.** **a**, **b** Variation of dielectric constant as a function of volume fraction of nanoparticles for (**a**) PVDF-TiO$_2$ nanocomposites and (**b**) PEI-TiO$_2$ nanocomposites under different temperatures. **c** The discharged energy density and energy efficiency at the breakdown strength for PVDF-TiO$_2$ nanocomposites with varying volume fractions of TiO$_2$ nanoparticles. **d** Variation of piezoelectric coefficient ($d_{33}$) as a function of volume fraction of nanoparticles in PVDF-TiO$_2$ nanocomposites, where the piezoelectric coefficients were tested after being poled under poling electric field of 150 kV/mm.

The piezoelectric charge coefficient $d_{33}$, another polarity-related property, of the PVDF-TiO$_2$ nanocomposites shows the same trend as the dielectric constant. The $d_{33}$ also reaches a peak value of −30 pC/N at about 0.35 vol.% of TiO$_2$ nanoparticles, with an enhancement of about 100% over the pristine PVDF polymer (Fig. 4d), leading to a much improved piezoelectric voltage coefficient $g_{33}$ of 360 mV.m/N and $d_{33} \times g_{33}$ product of 10.8 pm$^2$/N. The significantly improved $d_{33} \times g_{33}$ product is conducive to higher electrical energy output under dynamic force (Supplementary Fig. 21), which is expected to greatly benefit the applications of polymer nanocomposites in piezoelectric energy harvesting and flexible sensing. Of particular interest is that the remarkable enhancements in dielectric and piezoelectric properties of the dilute nanocomposites are achieved without sacrificing the mechanical stretchability, flexibility and transparency (Supplementary Fig. 22).

**Polar interfacial layer boosting ultra-high dielectric energy density**

These exceptional enhancements observed in the nanocomposites are the aggregate results from the two interfacial polar layers, whereas the individual contribution from each layer is indistinguishable. To further characterize their contributions, we prepared nanolaminates in which a PVDF thin film is confined between two Al$_2$O$_3$ nanolayers (Fig. 5a). The nanolaminated Al$_2$O$_3$/PVDF/Al$_2$O$_3$ nanocomposite exhibits two asymmetric PVDF-Al$_2$O$_3$ interfaces. The bottom active interface was first created by coating a PVDF thin film on the surface of the Al$_2$O$_3$ layer via the same solution-based process, forming the interfacial polar layers. The top Al$_2$O$_3$ layer was deposited onto the as-formed PVDF thin film via atomic layer deposition at a temperature far below the melting temperature of PVDF, which avoids the formation of an active interface

(Methods). Al$_2$O$_3$ was chosen to fabricate the nanolaminated nanocomposite because high-quality ultra-thin Al$_2$O$_3$ nanolayer can be easily prepared by atomic layer deposition at a temperature of 100 °C, meanwhile, the PVDF-Al$_2$O$_3$ nanocomposites with Al$_2$O$_3$ nanoparticles also exhibit a similar overall dielectric enhancement trend to the PVDF-TiO$_2$ nanocomposites (Supplementary Fig. 23). As shown in Fig. 5b, the $\varepsilon_r$ of the nanolaminated Al$_2$O$_3$/PVDF/Al$_2$O$_3$ nanocomposite exhibits a strong correlation with the PVDF film thickness (Supplementary Fig. 24), and increases with reducing the PVDF film thickness. Owing to the influence of the interfacial polar layers, the nanolaminated nanocomposites show a higher $\varepsilon_r$ than both Al$_2$O$_3$ ($\varepsilon_r$ ~ 9.8) and bulk PVDF ($\varepsilon_r$ ~ 7.8). The nanolaminated nanocomposite can be considered as a series connection of two Al$_2$O$_3$ films and PVDF film, therefore, the local $\varepsilon_r$ of the interfacial PVDF layers can be calculated based on the series law of capacitors (Supplementary Fig. 25). As shown in Fig. 5c, the $\varepsilon_r$ of PVDF layer within 10 nm from the Al$_2$O$_3$ surface, corresponding to the inner bound polar layer, boosts to a high value of 38.7, almost five times higher than its bulk value; the $\varepsilon_r$ of PVDF layer within the range between 10 and 300 nm away from the Al$_2$O$_3$ surface, corresponding to the outer polar layer, has an $\varepsilon_r$ of 13.2; while the PVDF beyond the 400 nm from the Al$_2$O$_3$ surface, exceeding the function range of the interfacial polar layers, has an $\varepsilon_r$ of 8, approaching to the value of bulk PVDF. It is of particular interest that the 10 nm-thick PVDF film also possesses excellent electric breakdown endurance due to the high density of polymer segments in the strongly adsorbed bound polymer layer, leading to an ultra-high breakdown strength greater than 1000 kV/mm for the nanolaminated nanocomposite (Supplementary Fig. 26). As a result, the Al$_2$O$_3$ (28 nm)/PVDF (10 nm)/Al$_2$O$_3$ (28 nm) nanolaminated nanocomposite containing the inner bound polar

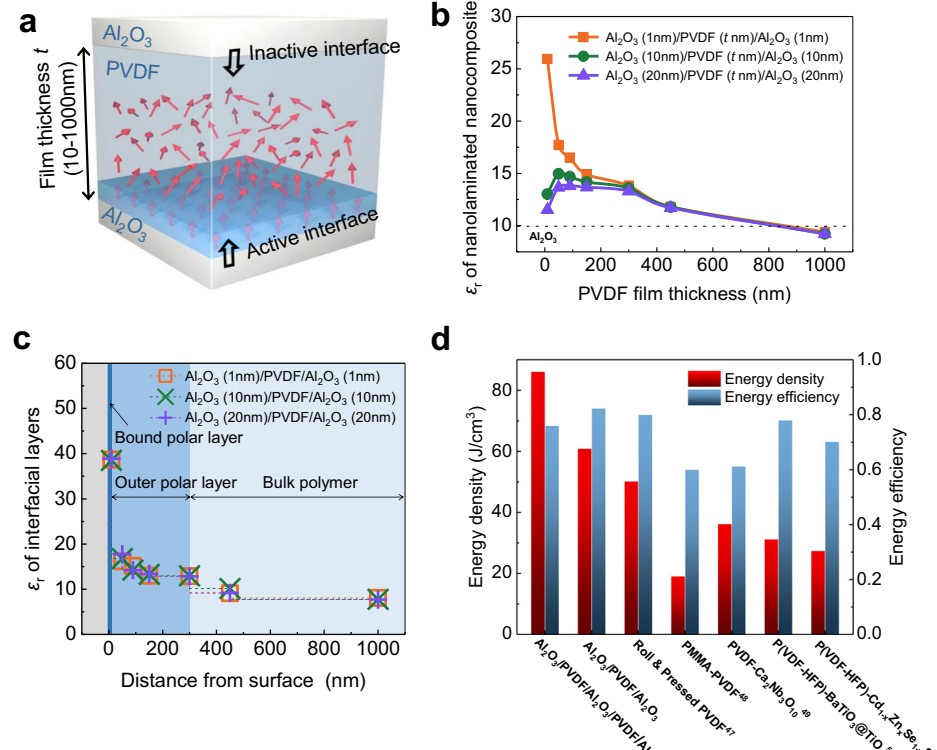

**Fig. 5 | The inner bound polar layer boosting significant interfacial effect.**
**a** Schematic illustration of the nanolaminated PVDF-Al$_2$O$_3$ nanocomposite with asymmetric active-inactive interfaces, the red arrows indicate the dipoles.
**b** Variation of the dielectric constant of nanolaminated Al$_2$O$_3$/PVDF($t$ nm)/Al$_2$O$_3$ nanocomposite with varying PVDF film thickness of $t$ nm. The thickness of both the top and bottom Al$_2$O$_3$ layers is controlled at 1, 10 and 20 nm. **c** The calculated distance-dependent dielectric constant of interfacial PVDF layers with varying distances from the bottom Al$_2$O$_3$ surface. **d** Comparison between dielectric energy storage densities and energy efficiencies for the nanolaminated nanocomposite and state-of-the-art polymer-based dielectrics reported in previous studies.

PVDF layer has an energy storage density of 61 J/cm$^3$. Notably, the Al$_2$O$_3$ (15 nm)/PVDF (10 nm)/Al$_2$O$_3$ (15 nm)/PVDF (10 nm)/Al$_2$O$_3$ (15 nm) nanolaminated nanocomposite containing two inner bound polar PVDF layers exhibits an ultra-high dielectric energy density of 86 J/cm$^3$, which is far beyond the values so far reported in the dielectric polymers and nanocomposites, while possessing a satisfactory energy efficiency of 76%, being comparable or even superior to most dielectric polymers and nanocomposites[47–51] (Fig. 5d and Supplementary Fig. 26).

In summary, we directly observe the presence of an interfacial polymer composed of high polarity bilayer surrounding the nanoparticles, being responsible for the substantially enhanced polarity-related properties at an ultralow nanoparticle addition in the nanocomposites, such as dielectric and piezoelectric properties without sacrificing the flexibility and transparency of the pristine polymer. By constructing layered polymer nanocomposites to fully exploit the contribution from the inner bound polar layer, ultrahigh dielectric energy storage density of the nanocomposites has been achieved. Unraveling the bilayer interfacial polymers and their impacts on the nanocomposite properties is essential for understanding the fundamental interfacial phenomena in polymer nanocomposites, which will provide great opportunities for design of new polymer nanocomposites with tailored polarity properties, being potential for high capacity dielectric energy storage and electrocaloric solid cooling, high performance piezoelectric energy harvesting, and emerging flexible and transparent piezoelectric sensing, to name a few.

## Methods

### Fabrication of polymer nanocomposites

Polymer nanocomposite film with nanoparticles protruded out was prepared on the silicon wafer substrate for local interfacial characterization (Fig. 1a). The TiO$_2$ nanoparticles with an average diameter of about 60 nm and PVDF (Arkema, Kynar Flex 2801) were dissolved into N,N-Dimethylformamide (DMF) at a concentration of 2% w/v and stirred for 24 h for homogeneous mixing. Silicon wafer with platinum top layer was obtained from Hefei Kejing Materials Technology, and was cleaned by ultrasonic bath. A pure PVDF polymer layer was first spin-coated onto silicon wafer substrates to eliminate the possible surface effect from the silicon substrate. Afterward, the PVDF-TiO$_2$ nanocomposite film was spin-coated onto the substrates with an as-coated PVDF layer at the spin speed of 8000 r/min, and then dried in a vacuum oven at 50 °C for 12 h.

Free-standing polymer nanocomposite films with a thickness of around ~10 µm were prepared for measuring their properties. The PVDF-TiO$_2$ nanocomposite mixture with PVDF/DMF solution concentration of 10% w/v was cast into films on glass substrates with a scraper. The films were then dried at 50 °C for 12 h in vacuum for the complete evaporation of solvents, after which, the flexible free-standing films were peeled from substrates. Free-standing nanocomposite films including PMMA-TiO$_2$, PEI-TiO$_2$, PVDF-Al$_2$O$_3$ were prepared with the same procedures, the Al$_2$O$_3$ nanoparticles have an average diameter of about 60 nm.

For the fabrication of layered Al$_2$O$_3$/PVDF/Al$_2$O$_3$ nanocomposites, the bottom Al$_2$O$_3$ layer was deposited at 100 °C onto the silicon wafer with platinum electrode using atomic layer deposition system (GEM-Star XT, Arradiance Inc., USA) with the Trimethylaluminum as precursor. The middle PVDF film was spin-coated onto the Al$_2$O$_3$ layer, and then dried in a vacuum oven at 50 °C for 12 h. The top Al$_2$O$_3$ layer was deposited onto the as-formed PVDF thin film also via atomic layer deposition at 100 °C. The thickness of both the bottom and top Al$_2$O$_3$ layers was controlled at about 1, 10 and 20 nm by controlling the deposition time duration, and the thickness of PVDF film was

controlled at about 10, 50, 90, 150, 300, 450 and 1000 nm by controlling the spin speed of coating.

## Structural characterization

The morphology of PVDF-TiO$_2$ nanocomposite was characterized with scanning electron microscopy (JSM-7610FPlus, JEOL, Japan). The TEM observation and the associated EDX mapping were conducted using a FEI Titan Cubed Themis G2 300 microscope with a spherical aberration corrector. AFM-IR measurements were implemented using a NanoIR3 (Bruker, USA). Contact-mode Si tips coated with gold of spring constant 0.07-0.4 N/m (Bruker, USA) were used. The polarized IR laser was set at 840 and 766 cm$^{-1}$ during the AFM-IR scanning.

## Molecular dynamic simulations

The initial single chain of PVDF was prepared by Materials studios (MS) 8.0 (20 monomers in each chain). The simulation model of PVDF was generated by the Amorphous cell module with the density of 1.77 g/cm$^3$. The final structure consists of 200 random chains with the box size of 5 nm × 5 nm × 20 nm. TiO$_2$ plate was perpendicular to z axis and the x-y dimensions were fixed to 4.8 nm × 4.8 nm.

We combine two kinds of force fields to describe the simulation system. The PVDF was simulated with the CVFF[52] force field. The UFF[53] force field was applied to parameterize the TiO$_2$ and to describe the interactions between TiO$_2$ and PVDF. Periodic boundary conditions were applied to all directions. Long-range correction was used to calculate Columbic interactions beyond 12 Å cutoff. Van der Waals interaction was also calculated with 12 Å cutoff. The temperature, T, was kept constant through the Nose-Hoover thermostat algorithm[54]. A time step of 0.1 femtosecond was used in all simulations. All molecular dynamics simulations were performed by the LAMMPS software package[55]. VMD software was used for visualization and post processing of simulation data[56].

## Phase-field simulations

The potential distribution around the interfaces can be obtained by solving the Boltzmann-Poisson equation[39],

$$\nabla^2 \psi(\mathbf{r}) = -q\varepsilon^{-1} \sum_i z_i n_i(\infty) \exp\left(-\frac{z_i q \psi(\mathbf{r})}{k_B T}\right) \qquad (1)$$

where $\psi(\mathbf{r})$, $q$, $\varepsilon$, $z_i$, $n_i$, $k_B$, and $T$ represent the local electrical potential, unit charge, dielectric constant, the dipole moment of dipole specie $i$, the concentration of dipole specie $i$, the Boltzmann constant and temperature, respectively. In this simulation, the diameter of nanoparticles was set to 60 nm, the electric potential at particle surface was set to 100 mV, the concentration of dipole was set to 10$^{13}$ cm$^{-3}$, respectively.

## Measurements of dielectric, piezoelectric and mechanical properties

Copper electrodes (4 mm in diameter) were coated onto the free-standing nanocomposites, and platinum electrodes (300 μm in diameter) were coated on the nanolaminated nanocomposites, for the measurements of dielectric and energy storage properties. Dielectric constants were measured with an impedance analyzer (HP 4294 A, Agilent, USA) with a broad frequency range from 10$^2$ to 10$^7$ Hz at 1 Vrms. Unipolar displacement-electric field (D-E) hysteresis loops (using a triangle waveform with a frequency of 100 Hz) measurements were measured by a multiferroic ferroelectric test system (Premier II, Radiant Technologies, Inc., USA) at room temperature. Electric breakdown tests were performed with a withstand voltage test system at a ramping rate of 200 V/s and 2 V/s for free-standing nanocomposite films and nanolaminated nanocomposite, respectively.

For the piezoelectric tests, copper electrodes (1 cm in diameter and 50 nm in thickness) were coated on both sides of nanocomposite film. The polymer nanocomposite films were first annealed at 140 °C for 24 h and then poled in silicone oil under the poling electric field of 150 kV/mm for 2 h. Then the film sample was held in place sandwiched by two rod probes (1 cm in diameter). The probes and electrodes with the same diameter were overlapped exactly. Dynamic force was applied on the upper probe during the test while the other was fixed, an electrometer (6517B, Keithley, USA) and a force sensor (YLK-20N, Elecall Electric, China) were employed for the detection of charge and force on the sample, respectively. The $d_{33}$ of polymer nanocomposite films was obtained by $d_{33} = Q/F$, where $Q$ is the charge measured by the electrometer and the $F$ is the applied force detected by the force sensor. The open-circuit voltages of the dilute nanocomposites were detected by an electrometer (6517B, Keithley, USA) under the periodic pressing force of 10 N. The stress-strain curves were tested with a universal material testing machine (C44, MTS SYSTEMS CO., LTD, USA).

## Data availability

The data that support the findings of this study are available within the article and its Supplementary Information files, and also from the corresponding author upon request.

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

## Acknowledgements

C.-W.N. was supported by the Basic Science Center Program of the National Natural Science Foundation of China (Grant No. 52388201). X.Z. was supported by the National Natural Science Foundation of China (Grant No.52222205, 52072280). X.Z. was supported by the National Key Research & Development Program (Grant No. 2021YFB3800603).

## Author contributions

X.Z. and C.-W.N. conceived and designed the research; X.Li, X.Z. and Y.J. prepared the polymer nanocomposites samples; S.H. and Y.S. collected the AFM-IR data, X.Li, Y.J. and Y.X. performed the structural characterization, PFM and dielectric measurements; J.W., Z.S. and X.Z. performed the phase-field simulations; Y.Y. and Y.L. performed the MD simulations; F.Z. and C.M. performed the HADDF characterization; X.Li, B.L. and X.Liu performed the piezoelectric measurements; X.Z., C.-W.N. and S.Z. analyzed the data and wrote the manuscript with input from all authors.

## Competing interests

The authors declare no competing interests.
