## [Peer Review File · Nature Communications]

Unraveling bilayer interfacial features and their effects in polar polymer nanocompositesReviewers' Comments:

Reviewer #1:

Remarks to the Author:

Very nice paper considering the interface on properties. This is considered widely, but the approach is of interest and detailed analysis.

English is generally ok but could be generally improved.

e.g. . "by bulging half of the nanoparticles ." bulging and bulge is an unusual term - "protrude" better

Avoid use of "etc." - vague.

Fonts are very small in Fig 1 - especially 1d and Figure 1d is an important image. This needs to be larger and more clear as it underpins the concept.

I also like supplementary Fig 1 -could it fit in the main paper body?

....since the previous interface model assumptions. [references needed here]

Fig 2 a - squares indicating the areas of interest and too hard to see - need to be thicker.

Figure 3c should indicate the d in the figure to help the reader (e.g. arrow between points with d) - it is easy to be confused and think it is size rather than spacing as not defined.

High d_{33} . g_{33} is stated to be "conducive to higher voltage output under dynamic force.." - a high g_{33} relates to high V (eg. sensor) while high d_{33} . g_{33} is high electrical energy output under dynamic force (e.g. harvester)

Fig 5a - again an important image and use of small fonts, also need to include an indication of typical scale or range.

5c - symbols so similar it is difficult to see

Composites have been examined with quite a low contrast between the filler particle and matrix - are there any issues when the contrast in properties between filler and matrix is larger e.g. ferroelectric fillers with high permittivity?

E.g. Breakdown in the Case for Materials with Giant Permittivity?

ACS Energy Lett. 2017, 2, 10, 2264

Chris Bowen

Reviewer #2:

Remarks to the Author:

This paper has data demonstrating the polarizability of PVDF near interfaces resulting in a significant increase in permittivity in both bulk and laminated composites. They are using IR-AFM to show the more polar phase of PVDF in the interface region. This leads to a higher permittivity than would be calculated from a rule of mixtures (for example). This has been observed in a few systems, and this work has directly measured the interfacial polarity. The authors have then used this knowledge to

make nanolaminates that have very high energy density.

The authors may want to look at these other AFM based papers that directly measure properties in the interface.

El Khoury, D.; Arinero, R.; Laurentie, J.; Bechelany, M.; Ramonda, M.; Castellon, J. Electrostatic force microscopy for the accurate characterization of interphases in nanocomposites. *J. Nanotechnology* 2018, 9, 2999–3012.

<https://doi.org/10.1021/acsaelm.2c01331>

<https://doi.org/10.1021/acs.macromol.8b01426>

A few questions:

Figure 1e is not convincing. It looks more like a full spherulite or crystalline region has different signal from the rest. It is much larger than 100nm, and while one could imagine the particle nucleated that particular region - it would suggest that a different phase, for example, of the polymer nucleated and grew. How do the authors know that this is just interfacial polymer? This is fundamental to the entire discussion. Is this nucleation from the surface and a different crystalline morphology (e.g. beta vs alpha phase) or is it a true interfacial region. It looks to me like a crystalline morphology. This does not take away from the overall results of the paper, but it would significantly change the discussion. For example, particles that are touching, may nucleate the alpha phase more easily due to the size of the nucleating agent.

I wonder if the paper should be split in 2. One paper showing the high energy density. A second on the technique and ability to see the morphology of the interfacial region?

Reviewer #3:

Remarks to the Author:

General comments:

In this study, by bulging half of the TiO₂ nanoparticles from a PVDF polar (ferroelectric polymer) matrix, two interfacial polymer layers (an internally bonded polar polymer layer of approximately 10 nm thick and a 100 nm thick interfacial polymer layer) were directly observed.

Furthermore, a combination of simulations and experiments revealed that both the nanoparticle-polymer interaction and the interparticle distance influence the formation of the interfacial polymer layer.

This paper further showed that this leads to exceptional enhancement of the polarity-related properties of dilute polymer nanocomposites.

In previous studies, surface potential mapping across the interface of nanoparticles and polymers enabled the detection of interfacial layers with high electrical polarization compared to the bulk polymer, but how these layers are spatially distributed has remained unsolved. That is, it was still a mystery what they were working for structure and properties of the nanocomposites.

This study clarified the effect of the decrease in the interparticle distance caused by increasing the loading of nanoparticles on the interfacial polymer, providing a comprehensive understanding of the interfacial effect in polymer nanocomposites. This point provides important knowledge not only in nanocomposite research but also in the field of adhesion of heterogeneous composite interfaces.

The methodology is described in enough detail that the experiment can be reproduced.

Specific comments:

Regarding my concern about this paper

In Fig. 2d: For TiO₂ nanoparticles with an average diameter of ~60 nm, the peak for 0.35 vol.% corresponds to a critical average interparticle distance of ~258 nm, what is the physical significance of this distance?

In this study the authors use TiO₂ nanoparticles with an average diameter of about 60 nm. Is there a result of changing the particle size?

In Fig. 1e: Is FTIR analysis possible from a localized area of the dense bonding layer having a higher density of polymer segments within the ~10 nm thick bonding layer than in the bulk polymer? Can all-trans TTTT conformation at the internally bonded polar polymer layer be detected?

In Fig. 3a: With a continuous increase in the content of TTTT polar structures, the amorphous content decreases, but why does it increase again as the potential on the particle surface increases?

Point-by-point Response Letter and List of Changes

We sincerely appreciate the reviewers' time and efforts for carefully reviewing our manuscript, providing valuable comments and suggestions. This uploaded manuscript has been revised (please see the **colored text** in the revised manuscript) according to the reviewers' advices. The point-by-point response (**in blue**) to the reviewers' comments and the main changes are listed in the following.

Reviewer #1:

Very nice paper considering the interface on properties. This is considered widely, but the approach is of interest and detailed analysis.

Response

We greatly thank the reviewer's positive evaluation of our manuscript. We have carefully addressed the reviewer's concerns, as discussed below.

1. English is generally ok but could be generally improved.e.g. . "by bulging half of the nanoparticles ." bulging and bulge is an unusual term - "protrude" better, avoid use of "etc." - vague.

Response

Thanks for the good comments. English of this manuscript has been polished, according to the reviewer's excellent suggestions, replacing "bulging and bulge" with "protruding and protrude", as well as avoiding the use of "etc".

2. Fonts are very small in Fig 1 - especially 1d and Figure 1d is an important image. This needs to be larger and more clear as it underpins the concept.

Response

Thanks for the good suggestion. The fonts in Fig 1 have been enlarged to enhance readability. Please see Fig 1 (or Figure R1 below) in the revised manuscript.

Figure R1 (Fig 1) Direct observation of the interfacial polymer around a nanoparticle in nanocomposites.

3. I also like supplementary Fig 1 -could it fit in the main paper body?

Response

We thank the reviewer for this valuable comment. Based on the reviewer's suggestion, we have moved supplementary Fig 1 into the main paper as Fig 1a. Please see Fig 1 (or Figure R1 above) in the revised manuscript.

4.since the previous interface model assumptions. [references needed here]

Response

We appreciate the reviewer for careful reading of the manuscript and comment. Several references have been incorporated into this position. Please see the first line on page 7 of the revised manuscript.

5. Fig 2 a - squares indicating the areas of interest and too hard to see - need to be thicker.

Response

We appreciate the reviewer for careful reading of the manuscript. The squares in Fig 2a have been emphasized to enhance their distinctiveness. Please see Fig 2a (or Figure R2 below) in the revised manuscript.

Figure R2 (Fig 2a) Interfacial polar regions in nanocomposites.

6. Figure 3c should indicate the d in the figure to help the reader (e.g. arrow between points with d) - it is easy to be confused and think it is size rather than spacing as not defined.

Response

We appreciate the reviewer for careful reading of the manuscript and comment. An arrow has been added in Figure 3c to define the d as the interparticle spacing. Please see Fig 3c (or Figure R3 below) in the revised manuscript.

Figure R3 (Fig 3c) Phase-field simulations of the interfacial polar polymers.

7. High $d_{33} \cdot g_{33}$ is stated to be “conductive to higher voltage output under dynamic force..” - a high g_{33} relates to high V (eg. sensor) while high $d_{33} \cdot g_{33}$ is high electrical energy output under dynamic force (e.g. harvester)

Response

We appreciate the reviewer for pointing out this mistake. We have revised the description in the revised manuscript, which is also listed below. Please see page 12 of the revised manuscript.

“The significantly improved $d_{33} \times g_{33}$ product is conducive to higher electrical energy output under dynamic

force”

8. Fig 5a - again an important image and use of small fonts, also need to include an indication of typical scale or range. 5c - symbols so similar it is difficult to see.

Response

Thanks for the good suggestions. The fonts in Figure 5a have been enlarged to improve readability, and the thickness of the central polymer film (10-1000 nm) is provided as an indicator of the typical scale of the nano-laminate. Please see Fig 5a (or Figure R4a below) in the revised manuscript.

The symbols in Fig 5c are highly overlapping. To enhance their discriminability, we have changed the symbol type and color.

Figure R4 (Fig 5) The inner bound polar layer boosting significant interfacial effect.

9. Composites have been examined with quite a low contrast between the filler particle and matrix - are there any issues when the contrast in properties between filler and matrix is larger e.g. ferroelectric fillers with high permittivity?

E.g. Breakdown in the Case for Materials with Giant Permittivity?

ACS Energy Lett. 2017, 2, 10, 2264

Response

We appreciate the reviewer’s insightful comments. We do agree that the contrast in permittivity between the filler and matrix is of significance for polymer nanocomposites since the dielectric contrast at the

interfaces can induce concentration of electric field as discussed in ACS Energy Lett. 2017, 2, 10, 2264.

To explore the potential impact of permittivity contrast, we proceeded to fabricate PVDF-BaTiO₃ nanocomposites, wherein ferroelectric BaTiO₃ exhibits a higher permittivity compared to TiO₂. By comparing the properties of PVDF-BaTiO₃ and PVDF-TiO₂ nanocomposites with an identical particle diameter of 80 nm, we can gain insights into the effect of permittivity contrast. According to the FTIR results of the PVDF-BaTiO₃ nanocomposites, the ratios of absorbance intensities at 840 cm⁻¹/766 cm⁻¹ were calculated. As seen in Figure R5, the content of the polar conformation in PVDF-BaTiO₃ composite continuously increases with increasing the volume fraction of the BaTiO₃ nanoparticles up to about 0.7 vol.%, and then decreases due to the reduced interparticle distance. It is noted that both the PVDF-BaTiO₃ and PVDF-TiO₂ nanocomposites exhibit the peak of polar content at about 0.7 vol.% of nanoparticles. This suggests that both BaTiO₃ and TiO₂ nanoparticles induce comparable interfacial polar characteristics in PVDF nanocomposites.

We further measured the dielectric permittivity and breakdown strength of the PVDF nanocomposites with BaTiO₃ and TiO₂ nanoparticles. As shown in Figure R6, although the BaTiO₃ has an intrinsic higher permittivity and leads to stronger electric field concentration, the difference in the resultant permittivity between PVDF-BaTiO₃ and PVDF-TiO₂ nanocomposites at the volume fraction range of 0.1 - 1 vol.% is minimal. In addition, PVDF-BaTiO₃ and PVDF-TiO₂ nanocomposites demonstrate similar breakdown strength across the filler volume fraction range of 0.1 - 1 vol.%, exhibiting no significant decrease compared to that of pure PVDF. Therefore, it may be concluded that the primary factor responsible for the enhancement of nanocomposite permittivity within the ultralow volume fraction range of 0.1 - 1 vol.% is the presence of polar interfacial polymers, while the contributions from filler permittivity and permittivity contrast are minimal due to their low content and screen effect of the polar interfacial polymers.

The permittivity contrast between fillers and matrix would exert a more salient influence on the properties of nanocomposites at elevated volume fractions. As shown in Figure R6, the PVDF-BaTiO₃ nanocomposite demonstrates a significantly higher permittivity than the PVDF-TiO₂ nanocomposite in the filler volume fraction range above 2 vol.%, with this feature becoming more pronounced at higher nanoparticle volume fractions. Meanwhile, the breakdown strength of the PVDF-BaTiO₃ nanocomposite is found to be more severely compromised than that of the PVDF-TiO₂ nanocomposite. The increase in permittivity of nanocomposites due to permittivity contrast comes at the cost of a decrease in dielectric strength, as discussed in ACS Energy Lett. 2017, 2, 10, 2264.

Some discussion has been added in the revised manuscript by citing this reference, please see page 11 and 12, which is also listed in the following. In addition, the comparison of the polar conformations and dielectric constant between PVDF-BaTiO₃ and PVDF-TiO₂ nanocomposites has been added to Supplementary Fig 12. On Page 11:

“A similar dielectric enhancement is also observed in various nanocomposites with different polymer matrices such as polyetherimide (PEI) and PMMA (Fig. 4b, Supplementary Fig. 11), as well as in nanocomposites containing BaTiO₃ nanoparticles. Despite BaTiO₃ exhibiting higher ϵ_r than TiO₂, both PVDF-BaTiO₃ and PVDF-TiO₂ nanocomposites demonstrate comparable polar conformations and dielectric enhancement across the volume fraction range of 0.1 - 1 vol.% (Supplementary Fig. 12), indicating that the

intrinsic ϵ_r of the nanoparticles may have little impact on the formation of the polar interfacial layers (Supplementary Fig. 12).”

On Page 12:

“The increase of $D(\epsilon_r)$ in polymer nanocomposites typically leads to a reduction in E_b ⁴⁶, while the intrinsic rise in ϵ_r resulting from polar polymer conformations maintains a high level of E_b for the nanocomposites. As illustrated in Supplementary Fig. 17 and 18, incorporating 0.35 vol.% of TiO₂ nanoparticles produces over 40% enhancement in D , while simultaneously preserving a high E_b of 630 kV/mm.”

46. Roscow, J. I., Bowen, C. R. & Almond, D. P. Breakdown in the Case for Materials with Giant Permittivity? *ACS Energy Lett.* **2**, 2264–2269 (2017).

Figure R5 a, b, (a) FTIR spectra and (b) the calculated ratio of absorbance intensities (A_{840} and A_{766}) at 840 cm⁻¹ and 766 cm⁻¹ for PVDF-TiO₂ nanocomposites with different volume fractions of nanoparticles. c, d, (c) FTIR spectra and (d) the calculated ratio of absorbance intensities (A_{840} and A_{766}) at 840 cm⁻¹ and 766 cm⁻¹ for PVDF-BaTiO₃ nanocomposites with different volume fractions of nanoparticles.

Figure R6 Variation of (a) dielectric constant and (b) Failure probabilities of breakdown strength deduced from Weibull distribution as a function of volume fraction of nanoparticles for nanocomposites.

Reviewer #2:

This paper has data demonstrating the polarizability of PVDF near interfaces resulting in a significant increase in permittivity in both bulk and laminated composites. They are using IR-AFM to show the more polar phase of PVDF in the interface region. This leads to a higher permittivity than would be calculated from a rule of mixtures (for example). This has been observed in a few systems, and this work has directly measured the interfacial polarity. The authors have then used this knowledge to make nanolaminates that have very high energy density.

Response

We greatly thank the reviewer for the assessments. We have addressed all the issues, as discussed below.

1. The authors may want to look at these other AFM based papers that directly measure properties in the interface. El Khoury, D.; Arinero, R.; Laurentie, J.; Bechelany, M.; Ramonda, M.; Castellon, J. Electrostatic force microscopy for the accurate characterization of interphases in nanocomposites. *J. Nanotechnology* 2018, 9, 2999–3012;

<https://doi.org/10.1021/acsaelm.2c01331>; <https://doi.org/10.1021/acs.macromol.8b01426>

Response

We greatly appreciate the reviewer's kind reminder of these important references. We have thoroughly reviewed these aforementioned papers, and appropriately cited them in the revised manuscript as Ref. 29-31.

2. Figure 1e is not convincing. It looks more like a full spherulite or crystalline region has different signal from the rest. It is much larger than 100nm, and while one could imagine the particle nucleated that particular region - it would suggest that a different phase, for example, of the polymer nucleated and grew. How do the

authors know that this is just interfacial polymer? This is fundamental to the entire discussion. Is this nucleation from the surface and a different crystalline morphology (e.g. beta vs alpha phase) or is it a true interfacial region. It looks to me like a crystalline morphology. This does not take away from the overall results of the paper, but it would significantly change the discussion. For example, particles that are touching, may nucleate the alpha phase more easily due to the size of the nucleating agent.

Response

Thanks for the reviewer's insightful comment. We agree with the reviewer that the PVDF nanocomposites involve intricate crystalline behaviors, given that PVDF is a semicrystalline polymer. Here, we would like to point out that polar interfacial phenomena can also occur in amorphous polar polymer nanocomposites, such as polymethyl methacrylate (PMMA), even in the absence of crystalline structures. We have performed a systematic analysis of the polar conformations in the PMMA-TiO₂ nanocomposites. As shown in Figure R7 (or Supplementary Fig 7 & 8), the polar configuration of PMMA, characterized by the mutual cis conformation of ester group and methoxy group, reaches a peak value at ~0.35 vol.% TiO₂ nanoparticles, followed by a subsequent decline. This finding is consistent with the observation in PVDF-TiO₂ nanocomposites. Therefore, we think that it is the interfacial polymers with polar configurations induced by surface potential that exert an influence, rather than their crystalline behaviors. This applies to both crystalline and amorphous polar polymer nanocomposites.

Certainly, we admit that we cannot definitively rule out the presence of crystalline phases surrounding the nanoparticle from the FTIR mapping results of PVDF-TiO₂ nanocomposites, as the β phase is also a polar phase of PVDF based on its all-trans TTTT polar configuration. As the reviewer mentioned, there is a possibility that the as-formed interfacial PVDF polymer with polar configurations may undergo crystallization into interfacial polar crystalline phases. However, we believe that this does not contradict our discussion based on the interfacial polar polymer configurations induced by particle surface potential, due to the fact that the β phase of PVDF is also composed of the all-trans TTTT polar polymers.

Following the reviewer's comments, we have added a discussion in the revised manuscript to provide further clarification on this issue, specifically by examining the scenario of crystalline structures. Please see page 7 of the revised manuscript, which is also listed here:

“Based on the observations in PVDF and PMMA-based nanocomposites, it is evident that the interfacial polar polymers are present in both crystalline (PVDF) and amorphous (PMMA) polar polymer nanocomposites. For crystalline polymers, such as PVDF, the as-formed interfacial polar polymers may undergo crystallization, leading to the development of interfacial polar crystalline phases, especially in the case of the β phase of PVDF.”

Figure R7 The correlation between interfacial polar polymers and inter-particle distance in PMMA-TiO₂ nanocomposites. a, Schematic illustration of PMMA polymer chain with mutual cis conformation of ester and methoxy groups, the dipoles are marked with orange arrows. b, FTIR spectra of PMMA-TiO₂ nanocomposites with different volume fractions of nanoparticles. c, FTIR spectra in the region 1050-1300 cm⁻¹ with fitted component bands at 1175 cm⁻¹, 1193 cm⁻¹, 1242 cm⁻¹ and 1273 cm⁻¹. d, The calculated ratio of absorbance intensities between bands of 1242 cm⁻¹ and 1273 cm⁻¹ for nanocomposites with different volume fractions of nanoparticles. The cis and trans conformations of the ester group (C-C-O) are assigned to the 1242 cm⁻¹ and 1273 cm⁻¹ respectively. e, The calculated ratio of absorbance intensities between bands of 1193 cm⁻¹ and 1175 cm⁻¹ in nanocomposites with different volume fractions of nanoparticles. The cis and trans conformations of the methoxy group (C-O-C) are assigned to the 1193 cm⁻¹ and 1175 cm⁻¹, respectively.

3. I wonder if the paper should be split in 2. One paper showing the high energy density. A second on the technique and ability to see the morphology of the interfacial region?

Response

We greatly appreciate the reviewer's insightful comment and suggestion. The feedback from the reviewer has been invaluable in shaping our future research direction. The current version by combination of the high energy density with the interfacial characterization can provide a more comprehensive narrative. Indeed, the high energy density polymer nanolaminate composite serves as an excellent example to showcase the implication of this observed interfacial morphology. As per the reviewer's recommendation, we will focus

on enhancing the energy storage performance of polymer nanocomposites by leveraging the polar interfacial phenomena in our upcoming research.

Reviewer #3:

In this study, by bulging half of the TiO₂ nanoparticles from a PVDF polar (ferroelectric polymer) matrix, two interfacial polymer layers (an internally bonded polar polymer layer of approximately 10 nm thick and a 100 nm thick interfacial polymer layer) were directly observed.

Furthermore, a combination of simulations and experiments revealed that both the nanoparticle-polymer interaction and the interparticle distance influence the formation of the interfacial polymer layer.

This paper further showed that this leads to exceptional enhancement of the polarity-related properties of dilute polymer nanocomposites.

In previous studies, surface potential mapping across the interface of nanoparticles and polymers enabled the detection of interfacial layers with high electrical polarization compared to the bulk polymer, but how these layers are spatially distributed has remained unsolved. That is, it was still a mystery what they were working for structure and properties of the nanocomposites.

This study clarified the effect of the decrease in the interparticle distance caused by increasing the loading of nanoparticles on the interfacial polymer, providing a comprehensive understanding of the interfacial effect in polymer nanocomposites. This point provides important knowledge not only in nanocomposite research but also in the field of adhesion of heterogeneous composite interfaces.

The methodology is described in enough detail that the experiment can be reproduced.

Response

We greatly thank the reviewer's positive evaluation of our manuscript. We have carefully addressed the reviewer's concerns with extra experiment and discussion.

1. In Fig. 2d: For TiO₂ nanoparticles with an average diameter of ~60 nm, the peak for 0.35 vol.% corresponds to a critical average interparticle distance of ~258 nm, what is the physical significance of this distance?

Response

Thanks for the reviewer's insightful comment. As shown in Figure R8 (or Supplementary Fig 6), the interparticle distance (d) is calculated under the assumption of an ideally homogeneous dispersion of nanoparticles within the polymer matrix, which is mathematically expressed as $d = D \left[\left(\frac{6f_{\text{particle}}}{\pi} \right)^{-\frac{1}{3}} - 1 \right]$, where f_{particle} represents the value fraction of the nanoparticles and the $D = 60$ nm denotes the diameter of

nanoparticles. This 0.35 vol.% signifies a critical point over which the overall polar configuration content begins to decline in PVDF-TiO₂ nanocomposites, which is associated with the emergence of peak enhancement in contents of the polar polymer conformations. Thus the critical average interparticle distance of ~258 nm estimated from 0.35 vol.% could be roughly considered as the maximum functional range of interfacial polar polymer between the nanoparticles in the polymer matrix.

Figure R8 (Supplementary Fig 6) a, The schematic illustration of the interparticle distance between two neighboring nanoparticles b, The variation of interparticle distance with the volume fraction of nanoparticles in the polymer nanocomposite.

2. In this study the authors use TiO₂ nanoparticles with an average diameter of about 60 nm. Is there a result of changing the particle size?

Response

Thanks for the reviewer's insightful comment. To investigate the effect of particle size, we have prepared PVDF-TiO₂ nanocomposites featuring TiO₂ nanoparticles with a diameter of 40 nm and 80 nm. Based on the FTIR measurement of the nanocomposites (Figure R9), it can be observed that both 40 nm and 80 nm sized TiO₂ nanoparticles induce a peak enhancement of polar conformation in the nanocomposites, which is similar to that observed for 60 nm TiO₂ nanoparticles. The variations in dielectric constants of nanocomposites containing nanoparticles of different sizes are consistent with changes in polar content (Figure R10). Notably, the peak value of both the dielectric constant and polar content shifts to a higher volume fraction as the diameter of nanoparticles increases. Specifically, for 40 nm, 60 nm, and 80 nm

nanoparticles, the peak value of the polar content appears at about 0.2 vol%, 0.35 vol%, and 0.7 vol% respectively.

Based on reviewer’s suggestion, the discussion has been added to the revised manuscript and is also listed in the following, please see page 11 of the revised manuscript and newly added Supplementary Fig 13.

“It is observed that the diameter of the nanoparticles has a close correlation with the occurrence of peak values of ϵ_r , where larger particle sizes necessitate a higher volume fraction of nanoparticles to achieve the peak value of the ϵ_r . For instance, the required volume fraction for achieving the highest ϵ_r shifts from 0.2 vol.% to 0.7 vol.% as the diameter of the TiO_2 increases from 40 nm to 80 nm (Supplementary Fig. 13).”

Figure R9 The calculated ratio of absorbance intensities (A_{840} and A_{766}) at 840 cm⁻¹ and 766 cm⁻¹ in the nanocomposites with different volume fractions of TiO₂ nanoparticles.

Figure R10 (Supplementary Fig 13) Variation of dielectric constant as a function of volume fraction of TiO₂ nanoparticles with different diameters.

3. In Fig. 1e: Is FTIR analysis possible from a localized area of the dense bonding layer having a higher density of polymer segments within the ~10 nm thick bonding layer than in the bulk polymer? Can all-trans TTTT conformation at the internally bonded polar polymer layer be detected?

Response

Thanks for the reviewer's insightful comment. We are sorry that it is difficult to characterize the localized area within the 10 nm thick bonding layer directly using AFM-FTIR, as this technique has a minimal spatial resolution of 10-20 nm. Nevertheless, we can detect the existence of TTTT conformation via the surface phase signal of the protruded nanoparticles by the piezoresponse force microscopy (PFM) test.

According to the cross-sectional SEM images (Figure 1c), we can observe that the 10 nm thick bonding layer is attached to the nanoparticle surface protruding out of the polymer matrix. Therefore, the characterization of the protruded nanoparticles' surface enables us to characterize the 10 nm bonding layer.

It should be noted that the all-trans TTTT conformation of PVDF produces the dipoles pointing from F to H atom due to the electronegativity difference between these two atoms. Therefore, PFM can be utilized to characterize the phase signal of these dipoles and detect the presence of all-trans TTTT conformation. The surface PFM phase images of the protruded nanoparticles are shown in Figure 1g, the lateral PFM (L-PFM) characterizations were conducted along the in-plane x-direction and also the y-direction via 90° rotation of the sample, in both cases, the nanoparticle has a half-dark and half-bright contrast. In vertical PFM (V-PFM), on the other hand, the nanoparticle shows no contrast. By combining L-PFM and V-PFM phase images, we can manifest that the protruded nanoparticles are covered with oriented dipoles, which are aligned perpendicular to the nanoparticle surface as illustrated in Fig. 1e. In contrast, no phase signals can be detected from the pristine TiO₂ nanoparticle surface without the PVDF bound layer as shown in Supplementary Fig. 5.

The characterization of the presence of dipoles in the bound polymer layer confirms that the inner 10nm thick bonding PVDF layer with high density also exhibits the all-trans TTTT polar conformation, due to the fact that the formation of the dipoles is based on the aligned all-trans TTTT conformation of PVDF.

Some discussion has been added to provide further clarification on this issue, please see page 6 of the revised manuscript, which is also included below:

“The all-*trans* TTTT polar conformation of PVDF causes the dipoles to point from F to H atom due to the electronegativity difference. To investigate the dipole orientations, lateral and vertical piezoresponse force microscopy was employed (PFM, Supplementary Fig. 2 & 3)^{37,38}. As shown in Fig. 1g, the lateral PFM (L-PFM) characterizations were conducted along the in-plane x-direction and also the y-direction via 90° rotation of the sample, in both cases, the nanoparticle has a half-dark and half-bright contrast. In vertical PFM (V-PFM), however, the nanoparticle shows no contrast. By combining L-PFM and V-PFM phase images, the interface dipoles are determined to be aligned perpendicular to the nanoparticle surface as illustrated in Fig. 1e. In contrast, no phase signals can be detected from the pristine TiO₂ nanoparticle surface

without the PVDF bound layer (Supplementary Fig. 4), which confirms that the orientation signal is originated from the bound polymer layer. The characterization of the presence of dipoles in the bound polymer layer also confirms that the inner 10nm thick binding PVDF layer with high density exhibits a TTTT polar conformation.”

4. In Fig. 3a: With a continuous increase in the content of TTTT polar structures, the amorphous content decreases, but why does it increase again as the potential on the particle surface increases?

Response

Thanks for the reviewer’s insightful comment. In our simulation, we categorized the configurations of PVDF into four main types: TTTT, TTTG, TGTG and amorphous. During tallying process of each configuration type, we begin by considering the contents of TTTT, TTTG and TGTG configurations; while classifying all other configurations as amorphous. However, it is worth noting that there may be certain PVDF configurations beyond TTTT that exhibit greater stability than the TTTG and TGTG configurations under high electric potentials, which are counted as amorphous configurations. Consequently, this leads to an overall increase in amorphousness under high electric potentials.

We apologize for this confusion. To clarify this issue, we revised the classification of “Amorphous” as “Amorphous and other” in Fig 3a.

Reviewers' Comments:

Reviewer #1:

Remarks to the Author:

Very good response to referee comments and response to my feedback is good. The paper is improved and also easier to read.

Reviewer #2:

Remarks to the Author:

My comments have essentially been addresses though I think the beta phase formation deserves further investigation.

Reviewer #3:

Remarks to the Author:

Thank you for providing answers to my questions and doing the corrective actions accordingly.

I appreciate the effort of the authors.

I have read in detail the responses and they address the issues raised.

It appears much stronger and suitable for publication.

Point-by-point Response Letter

We sincerely appreciate the reviewers' time and efforts for carefully reviewing our manuscript once again, and their positive recommendations. The following are our point-by-point response (in blue) to the reviewers' comments.

Reviewer #1:

Very good response to referee comments and response to my feedback is good. The paper is improved and also easier to read.

Response:

We thank the reviewer for his/her appreciations on the improvement of our revised manuscript, and also for his/her recommendation.

Reviewer #2:

My comments have essentially been addresses though I think the beta phase formation deserves further investigation.

Response:

We thank the reviewer for his/her appreciation on our responses and revised manuscript. Also, we would like to thank the reviewer for his/her valuable comment, and will further investigate the beta phase formation.

Reviewer #3:

Thank you for providing answers to my questions and doing the corrective actions accordingly. I appreciate the effort of the authors. I have read in detail the responses and they address the issues raised. It appears much stronger and suitable for publication.

Response:

We thank the reviewer for his/her appreciations on our responses and the improvement of our revised manuscript, and also for his/her recommendation.